# Mutant Cytochrome C as a Potential Detector of Superoxide Generation: Effect of Mutations on the Function and Properties

**DOI:** 10.3390/cells12182316

**Published:** 2023-09-19

**Authors:** Rita V. Chertkova, Ilya P. Oleynikov, Alexey A. Pakhomov, Roman V. Sudakov, Victor N. Orlov, Marina A. Semenova, Alexander M. Arutyunyan, Vasily V. Ptushenko, Mikhail P. Kirpichnikov, Dmitry A. Dolgikh, Tatiana V. Vygodina

**Affiliations:** 1Shemyakin-Ovchinnikov Institute of Bioorganic Chemistry, Russian Academy of Sciences, 117997 Moscow, Russia; alpah@mail.ru (A.A.P.); marinaapbch@mail.ru (M.A.S.); kirpichnikov@inbox.ru (M.P.K.); dolgikh@nmr.ru (D.A.D.); 2A.N. Belozersky Institute of Physico-Chemical Biology, M.V. Lomonosov Moscow State University, Leninskie gory 1, Bld. 40, 119992 Moscow, Russia; oleynikov.biophys@gmail.com (I.P.O.); sudakovromvlad@gmail.com (R.V.S.); orlovv@belozersky.msu.ru (V.N.O.); alarut@belozersky.msu.ru (A.M.A.); vygodina@belozersky.msu.ru (T.V.V.); 3N.M. Emanuel Institute of Biochemical Physics of the Russian Academy of Sciences, 119334 Moscow, Russia; 4Biology Department, M.V. Lomonosov Moscow State University, 119899 Moscow, Russia

**Keywords:** electron transport chain, mitochondrial cytochrome *c*, heme, cytochrome *c* oxidase, universal binding site, red Ω-loop of cytochrome *c*, electrostatic simulations, circular dichroism, dynamic light scattering, superoxide scavenging

## Abstract

Cytochrome c (CytC) is a single-electron carrier between complex bc1 and cytochrome c-oxidase (CcO) in the electron transport chain (ETC). It is also known as a good radical scavenger but its participation in electron flow through the ETC makes it impossible to use CytC as a radical sensor. To solve this problem, a series of mutants were constructed with substitutions of Lys residues in the universal binding site (UBS) which interact electrostatically with negatively charged Asp and Glu residues at the binding sites of CytC partners, bc1 complex and CcO. The aim of this study was to select a mutant that had lost its function as an electron carrier in the ETC, retaining the structure and ability to quench radicals. It was shown that a mutant CytC with substitutions of five (8Mut) and four (5Mut) Lys residues in the UBS was almost inactive toward CcO. However, all mutant proteins kept their antioxidant activity sufficiently with respect to the superoxide radical. Mutations shifted the dipole moment of the CytC molecule due to seriously changed electrostatics on the surface of the protein. In addition, a decrease in the redox potential of the protein as revealed by the redox titrations of 8Mut was detected. Nevertheless, the CD spectrum and dynamic light scattering suggested no significant changes in the secondary structure or aggregation of the molecules of CytC 8Mut. Thus, a variant 8Mut with multiple mutations in the UBS which lost its ability to electron transfer and saved most of its physico-chemical properties can be effectively used as a detector of superoxide generation both in mitochondria and in other systems.

## 1. Introduction

The canonical function of small globular protein cytochrome *c* (CytC) is a single-electron transfer between ubiquinol:cytochrome *c* oxidoreductase (complex bc1) and ferrocytochrome *c*:oxygen oxidoreductase (CcO) in the electron transfer chain (ETC) located in the inner mitochondrial membrane. The prosthetic group of CytC is hemoporfirin type *c* containing a central Fe atom that is tightly ligated by the nitrogen of His18 and more weakly by the sulfur of Met80 [1,2]. The formation of transient complexes between CytC and its redox partners on the ETC is necessary for electron transfer. According to modern concepts, the universal interaction site of CytC with the complex bc1 and CcO consists of a central hydrophobic domain and a surrounding electrostatic domain [3,4]. Electrostatic interactions acting at large distances determine the correct spatial orientation of the contacting proteins, while hydrophobic interactions are the main force stabilizing the complex. Electrostatic interactions are formed by the cluster of positively charged Lys residues on the surface of CytC (universal binding site, UBS) around the heme cavity and the negatively charged Asp and Glu residues on the subunits of the redox partners (CytC binding sites) [5,6].

To date, it is known that CytC has a relatively flexible structure, which allows it to adopt different conformations depending on the conditions, interactions, post-translational modifications, and localization, and, consequently, to exhibit different redox properties, functional activities, binding affinity, etc., see for ex. reviews [7,8,9]. Moreover, in addition to the conditions listed above, naturally occurring cytochrome c mutations also have an effect on the structure and function of this protein [10]. This CytC feature provides its multiple alternative functions in the cell, which have attracted considerable attention from researchers in recent years [7]. Another important CytC function is the apoptosis induction. Under the influence of various factors, CytC translocates across the mitochondrial outer membrane into the cytosol, where it either amplifies the external apoptotic signal or initiates the activation of the caspase cascade via the CytC-dependent apoptotic pathway by interacting with apoptotic protease-activating factor-1 (Apaf-1) during apoptosome assembly [11,12]. Permeabilization of the mitochondrial outer membrane is achieved through lipid peroxidation by the peroxidase-like activity of CytC in complex with cardiolipin (CL) [8,13]. As a result of the CytC–CL complex formation, CytC loses its native conformation and canonical functionality as an electron carrier in ETCs, while its peroxidase-like activity increases ten-fold.

CytC has the ability to neutralize free radicals, in particular the superoxide anion radical, one of the major reactive oxygen species (ROS). It is known that CytC can play a prominent role in the antioxidant defense of the cell due to its ability to scavenge superoxide [9,14].

In addition, a number of other important alternative functions of CytC are known—regulatory, signaling and catalytic activities: translocation of CytC to the nucleus and regulation of DNA repair processes under conditions of moderate damage [15]; translocation of CytC to the endoplasmic reticulum and interaction with the inositol 1,4,5-triphosphate receptor which leads to enhanced apoptotic signaling [16]; participation via Mia40-ALR(Erv) in the folding of dithiol proteins in IMS and oxidation by CcO provides the redox control of the process from the ETC [17,18], etc. It should be noted that the redox properties, functional, and conformational plasticity of CytC allow for its successful use as a basis for the construction of biosensors of various ligands [7], especially such biologically important ones as ROS and reactive nitrogen species (RNS), as well as in molecular bioelectronics [19].

We have previously constructed a mutant CytC variant with substitutions of five key Lys residues from the UBS to Glu residues, as well as three opposite Glu/Lys substitutions at positions 62, 69, and 90: K8E/K27E/E62K/E69K/K72E/K86E/K87E/E90K (8Mut). This mutant was designed to create a modified CytC that lacks electron transport activity but retains the ability to efficiently neutralize the superoxide anion radical. We have shown that in the mitoplast system, CytC 8Mut is inactive in the reaction with CcO and its ability to be reduced in the succinate:cytochrome *c* reductase reaction is no more than 3% compared to CytC WT [20]. Furthermore, in experiments in the submitochondrial particles system (SMP), the reduction of CytC 8Mut upon superoxide anion radical generation by complex I ETC was almost completely inhibited by superoxide dismutase. This allowed us to conclude that CytC 8Mut can be used for the direct quantitative analysis of superoxide radical generation by mitochondrial preparations with a greater efficiency and specificity than partially acetylated CytC, which is traditionally used for such measurements [20].

This article is devoted to the description of the design principles of the amino acid substitutions of the CytC variant 8Mut, as well as a detailed study of its functional activities and some physico-chemical properties. Two other mutant variants of CytC with substitutions, E69K/K72E (2Mut) [21] and K8E/E69K/K72E/K86E/K87E (5Mut), were selected from the extensive panel of intermediate mutant CytC obtained during the stepwise construction of CytC 8Mut. We have studied the effect of the substitutions, presumably in the UBS, on the spectral properties, physico-chemical properties, and antioxidant capacity of CytC. The aim of this work is to explore the properties of mutant forms of CytC in detail; in particular, to measure superoxide scavenging by under standard steady-state conditions, allowing the rates of the reactions to be estimated.

## 2. Materials and Methods

Components for the culture media and buffer solutions for chromatography and electrophoresis (AppliChem, Darmstadt, Germany), ampicillin, CytC from horse heart (Sigma, Schnelldorf, Germany), *Xho* I restriction endonuclease (Promega, Madison, WI, USA), *Bam*H I restriction endonuclease (New England Biolabs Inc., Ipswich, MA, USA), *Pfu*-DNA polymerase, and *T4*-DNA ligase (Fermentas, Pabrade, Lithuania) were used in this study. Distilled water was additionally purified on a Milli-Q system (Millipore, Burlington, MA, USA). TMPD (N,N,N′,N′-tetramethyl-p-phenylenediamine), L-ascorbic acid, potassium ferro- and ferricyanide, sodium dithionite, superoxide dismutase from bovine erythrocytes, xanthine oxidase from bovine milk, hypoxanthine, and hydrogen peroxide were from Sigma-Aldrich (Burlington, MA, USA). The concentration of CytC was determined from the difference absorption spectra (dithionite reduced vs. ferricyanide oxidized) using molar extinction coefficient ∆ε_550-540_nm = 18.7 mM^−1^cm^−1^. pH buffers and EDTA (ethylenediaminetetraacetic acid) were from Amresco (Solon, OH, USA). Water-soluble tetrazolium (WST-1) was from Dojindo Molecular Technologies (Tokyo, Japan). Dodecyl-maltoside of “Sol-Grade” type was from Anatrace (Maumee, OH, USA).

*1. Mutant Genes Construction.* Mutation of horse heart CytC gene within expression plasmid vector pBP (*CYC1/CYC3*) was performed via site-directed mutagenesis as recommended using the QuikChange Mutagenesis Kit (Stratagene, La Jolla, CA, USA) [21]. Oligonucleotide primers containing both substitutions were used to construct genes with two mutations, K86E/K87E and E69K/K72E. Mutated gene K86E/K87E was then used for further mutagenesis with corresponding primers to obtain CytC genes with two, five, and eight substitutions: E69K/K72E, K8E/E69K/K72E/K86E/K87E, and K8E/K27E/E62K/E69K/K72E/K86E/K87E/E90K. In addition, an extensive panel of mutant CytC genes containing from one to eight mutations was obtained. Intermediate CytC variants containing two, four, or six substitutions have been isolated and characterized previously [20,21]. The production of mutated DNA was analyzed electrophoretically in 1% agarose gel. The nucleotide sequences of the mutated CytC genes were determined using an ABI Prism 3100-Avant Genetic Analyzer automated sequencer (Applied Biosystems, Beverly, MA, USA). Selected mutant genes were cloned into expression vector pBP(CYC1/CYC3), modified for the expression of horse heart CytC [22].

*2. Expression of the Mutant Genes of CytC, Protein Isolation, and Purification*. Mutated genes were expressed in *E. coli* strain JM-109 grown at 37 °C for 22–24 h in vigorously stirred liquid medium SB containing ampicillin (0.2 mg/mL) [20,21]. The cells were harvested by centrifugation at 4000g (4 °C) for 20 min. The precipitate was resuspended in buffer (25 mM Na-Pi, pH 6.0) containing 1 mM NaN_3_ and frozen at –20 °C for 20–30 min.

The suspension was thawed, and the cells were disrupted using a French press (Spectronic Instruments, Inc., Irvine, CA, USA). The membranes were precipitated by centrifugation at 100,000× *g* for 20 min and the supernatants were collected. The protein was purified using an “AKTA FPLC” liquid chromatographic system (GE Healthcare, Chicago, IL, USA) as described previously [23,24]. The cell extracts prepared as described above were applied on an MP HS 10/10 cation-exchange column (BioRad, Hercules, CA, USA) equilibrated with the same buffer (25 mM Na-Pi, pH 6.0, 1 mM NaN_3_). The extracts containing mutant proteins with four or more substitutions were dialyzed against the same buffer before chromatography. CytC was eluted by the same buffer with a linear gradient of 1 M NaCl at the rate of 3 mL/min. The fractions eluted from MP HS were analyzed spectrophotometrically and by SDS-electrophoresis in 12% Tris-Tricine PAAG. The fractions containing cytochromes c were dialyzed against the buffer for absorption chromatography (10 mM Na-Pi, pH 7.0, 1 mM NaN_3_) and applied on a column with hydroxyapatite CHT-I (BioRad, Hercules, CA, USA). CytC was eluted in a 0.5 M Na-Pi (pH 7.0) gradient at the rate of 2 mL/min. The mutant CytC 8Mut was further purified via gel filtration on Superdex-200 10/300 column (GE Healthcare, Chicago, IL, USA) equilibrated with 50 mM Na-Pi (pH 7.2), 150 mM NaCl at the rate of 0.5 mL/min. The purification degree of CytC was estimated spectrophotometrically and by SDS-electrophoresis. The fractions containing 95%-pure proteins were combined and the samples were dialyzed twice against 10 mM ammonium carbonate, pH 7.9. The proteins were lyophilized on ALPHA I-5 and stored at –20 °C.

3. *Enzymatic activities.* The activity of CytC as an electron carrier was monitored with a covered Clark-type electrode as the rate of oxygen uptake by solubilized CcO using Oxytherm device (Hansatech, Norfolk, UK), in a thermostatted cell at 25 °C with permanent stirring. An amount of 5 mM ascorbate, 0.1 mM TMPD, and CytC at concentrations varying in a range from 1-4 μM were used as the oxidation substrate. The assays were performed in a basic medium (BM) containing 50 mM Hepes/Tris buffer, pH 7.6, 0.1 mM EDTA, 50 mM KCl. When measuring the oxidation rate with CcO, the buffer was also supplied with 0.05% dodecyl-maltoside (DM) to maintain CcO in the solubilized state.

4. *Spectrophotometric assays* were carried out in a standard semi-micro cuvette (Hellma, Müllheim, Germany) with blackened side walls, 10 mm light pathway, the slit width used was 1.5 nm. Absolute spectra of CytC were recorded in a BM buffer at a speed of 2 nm/s, in a double-beam spectrophotometer Cary 300 Bio (Varian, Palo Alto, CA, USA). Kinetic measurements of CytC radical scavenging were made on the spectrophotometer SLM Aminco DW-2000 (SLM Instruments, Wixom, MI, USA) in a dual-wavelength mode.

5. *Scavenging of superoxide radical* was monitored in 50 mM K-phosphate buffer, pH 7.5, with 0.1 mM EDTA as a kinetics of CytC reduction at 550 nm vs. 535 nm (λ reference) absorbance changes. The rate values were averaged from 3–4 recordings with indication of the deviations. Since preparations of CytC were at different states of oxidation, to standardize the conditions, substoichiometric amounts of ferricyanide were added to the samples and the completeness of oxidation was controlled spectrally [14].

6. *CD spectroscopy.* CD spectra were recorded in a BM with a Chiroscan CD spectrometer (Applied Photophysics, Surrey, UK) in a wavelength range from 190–260 nm at 25 °C in a quartz cuvette (Hellma analytics, Müllheim, Germany) with optical pass 0.01 mm and total volume ca 2 μL. The concentration of CytC in the sample was around 0.6 mM. CD values were expressed in terms of molar ellipticity (the difference in molecular extinction coefficients for left and right circularly polarized light, ∆ε_L-R_). The CD spectra were averaged from six recordings (at 0.5-nm/s) and smoothed with the instrument software. The CD spectra of the buffer were subtracted from the spectra of samples. The secondary structure analysis was performed on the basis of the CD spectra using CONTILL software (CDPro package, Colorado State University, Fort Collins, CO, USA).

7. *Dynamic light scattering* was recorded on the commercial device Zetasizer Nano ZS (Malvern Instruments, Malvern, Worcestershire, UK) with He-Ne laser (633 nm) in the cuvette with optical pass 10 mm and total volume ca 100 μL in a BM at CytC concentration 50μM. The curves were fitted using Dispersion Technology Software (DTS) version 4.2.

8. *Redox titrations* were made in a Cary 50 (Varian, Palo Alto, CA, USA) spectrophotometer. Titrations of CytC heme c were performed anaerobically in argon-flushed spectrophotometric cell with a Pt, Ag/AgCl electrode pair in a buffer 20 mM Tris-HCl (pH 7.5), 100 mM KCl supplied with 10 mM diaminodurol as the redox mediator. The medium contained 1 mM CytC. The electrode was calibrated before the experiment by a standard solution of 5 mM ferro/ferricyanide (1:1) in 1 M KCl assuming Em value of +420 mV [25]. Oxidative or reductive titrations were carried out stepwise by small additions of ferricyanide or ascorbate, respectively. The proportion of the reduced form of cyt *c*, [CytC_red_]/[CytC_total_], was determined from the difference absorption spectra (reduced minus oxidized) A_549_–A_556_ at different redox potential values and referred to the spectrum of the fully reduced enzyme obtained at the beginning of the measurements. The data were approximated by Henderson–Hasselbalch equation: [CytC_red_]/[CytC_total_] = 1/(1 + exp(E_h_–E_m_^0^)). The E_m_^0^ was used as an adjustable parameter and thus was determined from the experimental data.

9. *Electrostatic calculations* were performed using the Poisson–Boltzmann equation (PBE) solver DelPhi (V.4 Release 1.0) [26]. The high-resolution (1.9 Å) three-dimensional structure of horse heart CytC (the PDB code 1HRC) was used. The electrostatic effect of replacements was assessed by changing charges localized on the corresponding amino acid residues (−1 → 1 or 1 → −1). To assess the dielectric permittivity (ε) of the protein, the algorithm elaborated earlier [27] was used. Since the protein was small enough, we did not take into account its dielectric inhomogeneity; the mean value ε = 20 was used instead.

10. *Isoelectric surfaces* were built from electrostatic potentials calculated using the Pois-son–Boltzmann equation in the ProKSim program [28]. The following parameters were used for the calculations: ionic strength I = 100 mM, maximum radius of electro-static interaction A = 35 Å, temperature T = 300 K, dynamic viscosity of the medium v = 0.001004 kg/(m × s), pH 7.0, the step for calculating the potential was 1 Å, 1000 iterations of solving the Poisson–Boltzmann equation. The atomic partial charges were taken from the Charmm27 force field. For CytC WT, the crystal structure (PDB ID 1HRC) was used. The 3D structure of CytC mut8 was generated using PyMol software V. 2.4.1. (Schrödinger, Inc, New York, NY, USA) and before electrostatic calculations, the structures were optimized via energy minimization using GROMACS (V. 2020.1, 2023.1) software (Uppsala University, Sweden), similar to [29].

## 3. Results

### 3.1. Construction of Substitutions in the Universal Cytochrome C Binding Site to Obtain a Protein Devoid of Electron Transport Activity

The horse CytC molecule contains 19 Lys residues distributed on the protein surface, 10 to 12 of which form the UBS CytC and surround the heme cavity. The UBS plays a key role in the formation of the CytC reaction complexes required for efficient electron transfer in the ETC. Highly conserved Lys at positions 8, 13, 72, (79), and 86 are key residues and Lys 25, 27, 73, (79), and 88 are located on the periphery of the contact surface and participate in binding to a lesser extent [5,6,30,31]. Notably, some peripheral Lys residues, for instance, at positions 73 and 79, are sometimes also referred to as key UBS residues. Of interest, the role of Lys residues from this series is also decisive in the interaction of CytC with a wide variety of other partners of protein and non-protein nature-cytochrome *c*-peroxidase [32], cytochrome *b5* [33], Apaf-1 [34], CL [35], histone chaperones [36], neuroglobin [37] and others [7], which confirms the versatility of this CytC site.

In order to suppress the electron transport activity of CytC, we replaced the key positively charged Lys residues (except for Lys13 and Lys79) with Glu residues in the binding site with the redox partner - complexes bc1 and CcO of ETC. The Lys13 residue located between the Lys8, 27, 86, and 87 residues was left intact for the compensation of the significant local negative charge arising from the substitution of these Lys to Glu (Figure 1). Compensatory substitutions of Glu62, 69, and 90 residues for Lys residues were made to stabilize the protein structure. The location of compensatory substitutions was chosen in such a way that the site of the positively charged Lys CytC residues interacting with the Asp and Glu residues on the bc1 and CcO complexes was “shifted” from the heme cavity to the periphery due to a change in the dipole moment of the CytC molecule (Figure 2). The dipole moment plays a crucial role in the correct spatial orientation of the proteins, which increases their affinity to different partner molecules during the complex formation. In the case of CytC, the dipole moment arises from the cluster of negatively charged Glu residues on the side of the molecule opposite to the front (Figure 1). In order to achieve a more significant shift in the dipole moment, the Lys79 residue was not subjected to mutagenesis.

In addition, an intermediate mutant variant of CytC with E69K/K72E substitutions (CytC 2Mut) was obtained, with minimal substitutions in the UBS, but the dipole moment of the protein was changed due to the compensatory substitution of E69K. In CytC 5Mut, additional replacements K8E/K86E/K87E were introduced into the site of interaction with the ETC partners. CytC 8Mut was obtained by introducing an additional K27E substitution near the heme cavity and compensatory substitutions E62K/E90K. Thus, when designing mutant CytC, we introduced substitutions that destroy the interactions of cytochrome C with redox partners and change the landing site by changing the direction of the dipole moment of the protein, while maintaining the balance of the charges and the stability of the structure as a whole.

### 3.2. Functional Activities of Mutant Cytochrome C Variants

#### 3.2.1. Electron Transport Activity of Cytochrome C in Reaction with Cytochrome C Oxidase

In the course of studying the interaction of CytC and its mutant variants with the solubilized CcO complex, it was shown that the replacement of Lys residues causes a significant disruption of the main enzymatic function of CytC in the ETC—electron transfer to CcO. In the case of CytC 2Mut, the electron transport activity is reduced by 75% compared to CytC WT and is almost completely suppressed in the mutant variants of CytC 5Mut and 8Mut (Figure 3). These data are in consistence with the previous studies’ data on the respiratory activity of mutant CytC in the mitoplast system [20]. The oxygen consumption kinetics of solubilized CcO oxidizing CytC in the presence of tetramethyl-p-phenylenediamine (TMPD) and ascorbate are shown in Figure 3. For the convenience of visualization during the registration of respiration in the case of CytC WT, the concentration of CcO was reduced by five times. According to the data shown, in the case of CytC 2Mut (red line), as well as WT (black line), in the micromolar concentration range, the rate of oxygen consumption increases in proportion to the concentration of added CytC. The inset shows that a directly proportional dependence on the concentration of CytC 2Mut added to the medium is observed regardless of the presence of the mediator TMPD. This confirms that CytC is a direct electron donor for CcO. Reliably recorded oxygen consumption is observed only after the addition of TMPD to the medium, and the subsequent addition of the mutant CytC almost does not increase the rate of the process for the 5Mut and 8Mut mutant CytC variants. Upon reaching a concentration of 3 µM, the subsequent addition of the mutant CytC (fourth addition), instead of accelerating, causes the inhibition of the reaction, which is especially evident for CytC 5Mut. Thus, the uptake of oxygen CcO (oxidase reaction) with the participation of the mutant forms of CytC 5Mut and 8Mut (green and blue lines, respectively) proceeds mainly due to the oxidation of TMPD, and not mutant cytochromes. Therefore, we can conclude that there is an almost complete loss of their electron transport activity. The quantitative values of the electron transfer rates are given in Table 1 (see below).

#### 3.2.2. The Ability of Mutant Cytochromes C to Be Reduced by Superoxide Anion Radical

It was previously shown that mutant forms of CytC with reduced or almost completely eliminated electron transport activity (CytC 8Mut) can be successfully used to measure the generation of the superoxide anion radical in an SMP system [20]. However, the generation of the superoxide under physiological conditions depends on a large number of factors, whereas it is important to measure the effect of single or multiple mutations under controlled standard conditions. In this regard, herein, we used the hypoxanthine oxidation reaction catalyzed by xanthine oxidase, which is considered a standard method for inducing superoxide generation at a constant rate.

Figure 4 shows the reduction kinetics of CytC WT (black line) and the mutant forms CytC 2Mut (red line), CytC 5Mut (green line), and CytC 8Mut (blue line) by the superoxide produced in the oxidation of hypoxanthine by xanthine oxidase.

The concentration of CytC used in the measurement medium (4 µM) provides a rather extended linear portion of the kinetic curves, which makes it possible to determine the reaction rate. The decrease in the rate of reduction over time is associated with a decrease in the ratio of the concentrations of ferric and ferrous CytC. It should be noted that an increase in the initial concentration of CytC in the sample did not affect the rate of superoxide annihilation, but only increased the length of the linear part of the curve. Thus, under these conditions, the recorded reaction has a pseudo-zero order. In a separate experiment, it was shown that the reduction of CytC is completely inhibited by superoxide dismutase. The mean rates obtained and the standard deviations calculated from three to four repetitions are shown in Table 2. In parallel experiments, the rate of superoxide O_2_^−^ generation was measured using water-soluble tetrazolium WST-1, which is reduced by superoxide to formazan by accepting two electrons. The obtained value of 1.4 ± 0.09 uM O_2_^−^/min turned out to be quite close to the value of 1.63 ± 0.07 given in Table 2 for CytC WT.

### 3.3. Influence of Introduced Substitutions on the Redox Properties of Cytochromes C

In order to assess to what extent the observed changes in the CytC redox reactions can be associated with a possible change in its redox potential, we performed potentiometric titrations of CytC WT and CytC 8Mut. The measurements revealed that the standard cytochrome *E*_m_^0^ redox potential is reduced by 27 mV for CytC 8Mut compared to CytC WT (Figure 5). The direction of the change in the *E*_m_^0^ indicates that it can be caused by a change in the overall charge of the protein as a result of amino acid substitutions. Of note, complete charge compensation was not achieved and CytC 8Mut carries an additional negative charge in comparison to CytC WT.

Theoretical estimates were performed to understand what proportion of the observed *E*_m_^0^ shift is due to a change in the total protein charge. In the first approximation, the effect of each substitution on the *E*_m_^0^ of the CytC can be estimated using the known three-dimensional protein structure. However, this approach does not take into account the change in the mutant protein structure compared to the wild-type protein in the changed electric field of its own charges. This dielectric relaxation should reduce the change in the heme redox potential caused by the electrogenic amino acid substitutions. Hence, these calculations should give an upper estimate of changes in the *E*_m_^0^.

Indeed, we estimated the electric effect of substitutions in CytC 8Mut as −48 mV, i.e., almost 1.8 times greater in modulus than the experimentally measured change. However, it gives a reasonable upper estimate for the *E*_m_^0^ shift due to electrogenic substitutions. Table 3 also shows the estimation of the *E*_m_^0^ shift in two other CytC mutants based on the wild-type 1HRC structure. Hence, the changes for all the mutants we studied are small (on the order of kT) and are directed toward a decrease in the *E*_m_^0^ (i.e., an increase in the reducing properties) of CytC. The calculations were performed for the protein permittivity ε = 20, a value based on the previously developed algorithm for estimating the ε distribution inside the protein based on its spatial structure [27]. In most of the early works, the semicontinuum approach to protein electrostatics generally used significantly lower permittivities (ε = 4–6). For comparison, we also performed a calculation at ε = 5, which gave approximately twice as large values for the *E*_m_^0^ shift in all mutant proteins: −26 mV, −43 mV, and −86 mV for CytC 2Mut, 5Mut, and 8Mut, respectively.

Of note, the decrease in the *E*_m_^0^ of the CytC, i.e., the weakening of its oxidizing properties should lead to some decrease in the rate of its reduction by superoxide. According to the Marcus theory, the activation energy of the electron transfer reaction Δ*G*^#^ in the course of the redox reaction in a polar medium is related to the reaction energy Δ*G*^0^ as
Δ*G*^#^ = (*λ*_o_ + Δ*G*^0^)^2^/4*λ*_o_, (1)
where *λ*_o_ is the reorganization energy of the polar environment. Therefore, at relatively small (compared to the reorganization energy) values of the reaction energy (which can be considered an acceptable approximation in this case), the change in the free energy of the reaction is accompanied by a change in the free energy of activation that is approximately half as large:ΔΔ*G*^#^ ≈ ΔΔ*G*^0^/2 (2)

In the case of CytC 8Mut, its more negative redox potential should lead to an increase in Δ*G*^#^ by about 13.5 mV, which corresponds to a decrease in the reaction rate by ~40% (by a factor of 1.7). This is quite close to the experimentally observed slowdown in the reduction reaction of CytC and superoxide (see Table 2).

### 3.4. Effect of the Introduced Mutations on the Physico-Chemical Properties of Cytochromes C

#### 3.4.1. Absorption Spectra

The absorption spectra of the studied mutant CytC virtually did not differ from the spectra of CytC WT either in the position or in the magnitude of the peaks. However, small differences were observed in the ferric forms of proteins in the far-red region (Figure 6). The ferric form was obtained by adding substoichiometric amounts of ferric cyanide, since the original protein preparations were in different redox states. Ferric CytC has a weak (about 0.2 mM^−1^cm^−1^) peak with a charge transfer at 695–698 nm, reflecting the state of the Met80 bond with heme iron. It can be seen that the peak at 695–698 nm for recombinant cytochromes is slightly smaller (the maximum value is about 0.13 mM^−1^ cm^−1^) than for native CytC isolated from horse heart. Approximately the same picture is typical of other mutant cytochromes. Furthermore, all recombinant cytochromes have a new absorption peak at 653 nm, which is absent in native CytC from horse heart. For CytC 8Mut (red line), this peak is 0.35 mM^−1^ cm^−1^, which is two to three times higher than the peak at 695 nm, while in the case of CytC WT (blue line) and other mutants, their amplitudes are approximately equal. The peak at 653 nm is also present only in the ferric form and disappears when the protein is reduced (dotted red line).

#### 3.4.2. Circular Dichroism Spectroscopy

The CD spectra (normalized to the concentration of CytC in the sample, determined after the experiment) were recorded in the far UV (Figure 7), and the local minima were noted at 208 nm and 222 nm, which are characteristic of α-helical structures. The nature of the spectra indicates that most of the structure of cytochromes is represented by α-helices. At the same time, for the CytC 2Mut and 8Mut variants, the CD spectra are in consistence with the spectrum of CytC WT, which indicates that the secondary structure of CytC is generally retained. This suggests that the introduced amino acid substitutions do not cause significant changes in the composition of the secondary structure, at least in the CytC 2Mut and 8Mut mutants. The CD spectrum of the CytC 5Mut variant is characterized by the same bands (at 208 and 222 nm) typical of the α-helical secondary structure. However, the intensity of these bands is noticeably higher than that of CytC WT, and this difference is reproducible and repeated in independent experiments. We do not comprehend the exact reason for the observed differences in the CD spectra; however, they may be due to an error in determining the concentration of the 5Mut protein, which was based, as described in the Section 2, on the absorption at 550 nm. Nonetheless, if the concentration of all CytC preparations is determined by the absorption in the far UV region, as recommended in Kelly et al. [38], there is a slight spread in the CD spectra, but no significant differences were observed (inset in Figure 6). We decided to present both figures in the article, which indicate that CytC 5Mut, whose charge is compensated the least (5 K->E substitutions against only one E->K), has a rather rare phenomenon of a decrease in the molar extinction in the visible region. A change in the molar extinction usually causes a broadening of the band at 550 nm, which is not observed in this case. However, electrostatic disturbances near the heme cavity due to mutations can contribute, for example, to the appearance of a heme-free apoprotein. A more detailed study of this phenomenon was beyond the scope of this article.

Given the above, we also hypothesized that this observed difference may be due to an increase in the turbidity of the samples caused by particle aggregation, so it was decided to compare the samples by the dynamic light scattering they cause.

Analysis of the secondary structure composition of the mutant CytC variants revealed that in the case of the 2Mut and 8Mut mutants, the content of the α-helical structure slightly increases, mainly due to a decrease in the content of the β-structural elements (Table 4). Furthermore, in the case of the 5Mut mutant, a sharp, almost two-fold, increase in the content of the α-helical structure was observed.

#### 3.4.3. Dynamic Light Scattering Revealed a Tendency for Aggregation of the CytC 5Mut

Dynamic light scattering revealed the heterogeneity of the samples of the studied cytochromes. According to the data obtained in Figure 8 for the mutant CytC 2Mut and, especially, for 8Mut, the scattering distribution is quite close to the distribution of particles in the commercial CytC from Sigma. It is quite homogeneous and falls on particles 3–6 nm in size, while the CytC 5Mut variant, along with a peak at 5 nm, has a pronounced shoulder of about 9 nm, with approximately 15% of the scattering particles having a size of 10–15 nm.

Apparently, CytC WT exhibits a tendency for dimers formation, and as a result, about half of the particles have a size of 3 nm, corresponding to the monomeric state, and half are 6 nm. Hence, the commercial CytC (Sigma) was used as a control sample. It can be concluded that the introduced mutations cause a tendency for particle aggregation in the samples of 2Mut and 5Mut, which more evidently manifests itself in CytC 5Mut, whose charge is the more uncompensated. The data obtained are in consistence with the CD data: for CytC 5Mut, an increase in the amplitude of the CD spectrum is observed, which is absent in the spectra of other CytC variants.

## 4. Discussion

CytC is one of the most ancient redox proteins, originated in anaerobic photosynthetic bacteria at the dawn of evolution more than 3 billion years ago [39]. As mentioned in the introduction, CytC has a rather flexible structure that allows it to serve as an electron carrier in ETCs and to perform multiple alternative functions, including regulatory, catalytic, signaling, etc. [7]. Efficient electron transfer in the ETC is due to the formation of transient complexes between CytC and its redox partners—bc1 and CcO complexes. The formation of such complexes generally occurs in two steps. In the first step, long-range electrostatic interactions between a cluster of Lys residues (UBS) located around the CytC heme cavity and the sites of redox partners consisting of three to four negatively charged Glu and/or Asp residues are involved. In the second step, the complex is stabilized by the hydrophobic, ionic, and van der Waals interactions of a number of amino acid residues. In this case, the hemoporphyrin acquires the conformation and orientation necessary for electron transfer, and its edge shifts closest to the edge of hemoporphyrin *c*1 of the bc1 complex or the center of CuA of the CcO.

In order to solve the issue of creating a CytC capable of interacting with ROS, but devoid of electron transport activity, we constructed a number of mutant variants with amino acid residue substitutions in the universal CytC binding site. Out of all the key Lys residues for interaction with redox partners, five residues were replaced by the Glu residue (except K13, CytC variant 8Mut). In terms of intermediate mutant proteins obtained in the course of construction in this study, CytC 5Mut has four Lys residue substitutions (except for K13 and K27) and CytC 2Mut - one substitutions (K72E) in the UBS. Glu/Lys residue substitutions that are opposite in charge—E62K, E69K, and E90K in CytC 8Mut (Figure 1) and E69K CytC 5Mut and 2 Mut—serve to partially compensate for the total protein charge. In addition, it was assumed that in the final version of CytC 8Mut, the substitutions E62K, E69K, and E90K would contribute to the “shift” of positive charges from the region around the heme cavity to the periphery, which leads to a shift in the dipole moment of the molecule and, as a result, to inefficient interaction with the redox partners of the *bc_1_* and CcO complexes (Figure 2).

According to our results, the replacement of Lys residues from the UBS, which are key for interaction with the redox partners of CytC at the ETC, by Glu residues, which prevent effective electrostatic interactions of mutant CytC with CcO, as well as a shift in the dipole moment of the molecule, lead to the almost complete suppression of electron transport activity in mutant CytC 5Mut and 8Mut (Figure 3, Table 1). Furthermore, mutations introduced into the UBS practically do not affect the rate of reduction of mutant CytC by the superoxide anion radical (Figure 4, Table 2). It is evident that the rate determined using CytC 8Mut is reduced compared to that for CytC WT by no more than 50%. This finding can be explained by the fact that effective electron transfer from the superoxide anion radical does not require the formation of a transient complex, as is custom for electron transport during the interaction of CytC with ETC partners. Nevertheless, the observed decrease in the antioxidant activity of CytC 8Mut (as well as CytC 5Mut and 2Mut) can be partially explained by the introduction of negatively charged Glu residues in the close proximity of the heme cavity, which create electrostatic obstacles for the reaction with the negatively charged anion of the superoxide radical. Moreover, due to the introduction of uncompensated negative charges into the molecule, the total positive charge of CytC decreases, which can reduce the rate of reaction with the superoxide radical as well.

A comparative analysis of the obtained data made it possible to conclude that if the introduced mutations in the UBS drastically impair the ability of CytC to transfer electrons to CcO, then their function of antioxidant protection, expressed in the ability to neutralize the superoxide anion radical, is largely preserved. Thus, the reduction rate of CytC 8Mut with superoxide decreases by no more than two times compared with CytC WT, while its activity in the electron transport chain on CcO is almost completely suppressed (a similar picture is also characteristic of CytC 5Mut). The data obtained allow us to conclude that the inhibition of the electron transport function in mutant cytochrome c is apparently associated with difficulties in the formation of a reactive complex with CcO caused by the replacement of key Lys residues by oppositely charged Glu residues and a shift in the dipole moment of the molecule. At the same time, the ability of mutant CytC to recover from the superoxide anion radical is largely retained. It should be noted that our data on the interaction of the mutant CytC 8Mut with CcO are in consistence with the results of testing the electron transport activity of this protein in the mitoplast system, as well as with the results of experiments on the quantitative determination of the rate of generation of the superoxide anion radical in the SMP system using CytC 8Mut [20].

The performed potentiometric titrations revealed a significant negative shift (by −27 mV) of the half-reduction potential (*E*_m_^0^) of CytC 8Mut compared to CytC WT (Figure 5). Theoretical estimates of this shift, performed using the Poisson–Boltzmann equation, show that the decrease in the *E*_m_^0^ is most likely due to the electrostatic effect of additional negative charges introduced into the protein on heme. In turn, this shift is probably responsible for some decrease in the activity of CytC in the superoxide oxidation reaction. This allows us to suggest that the introduction of substitutions into CytC, which will more compensate for changes in the protein charge, can make it possible to obtain a mutant form of CytC that is not inferior to CytC WT in terms of the activity of the superoxide oxidase reaction, but does not participate in interactions with the components of the respiratory chain.

Our data indicate that substitutions of Lys amino acid residues from the UBS and a shift in the dipole moment of the molecule lead to the disruption of the main function of CytC as an electron transporter in the ETC (Figure 3); however, they have almost no effect on the secondary structure of CytC 8Mut (Table 4). The predominance of negative charges in the mutant CytC 5Mut (four K/E substitutions versus one E/K) leads to some structural changes in the protein, expressed in a significant increase in the amplitude of the CD spectrum in comparison with the CD spectra of CytC 2Mut, 8Mut, and CytC WT, which sufficiently coincide (Figure 7), as well as the aggregation tendency of CytC 5Mut, revealed by the dynamic light scattering (Figure 8). The aggregation tendency in comparison with CytC WT is also characteristic of CytC 2Mut; however, the degree of aggregation of CytC 5Mut molecules is much higher. Moreover, a decrease in the intensity of the peak at 695–698 nm, which reflects the state of the Met80 bond with heme iron, may indicate some increase in the lability of the Fe…S(Met80) bond. It should be noted that 8Mut showed no tendency to aggregation at all, and its particle size, on average, 3 nm, corresponded to the native state of CytC.

Of note, it was previously shown by Chertkova et al. [40,41], using the methods of CD and IR spectroscopy, as well as resonance Raman spectroscopy (RRS) and surface-enhanced Raman spectroscopy (SERS), that site-directed mutations in the red Ω-loop (70–85) lead to a significant loss of the heme ability to adopt a planar conformation, which is necessary for optimal electron transfer. This was due to a decrease in the conformational lability of the Ω-loop of mutant forms of CytC and, as a result, an increase in the rigidity of the hemoporphyrin microenvironment. Furthermore, according to RRS/SERS spectroscopy data, mutations in the UBS CytC lead to the opposite effect. Both in the free state and bound to CL-containing liposomes, CytC 8Mut showed an increase in the probability of a planar heme conformation compared to WT CytC, which indicates a closer location of the Fe atom to the heme plane in this protein [42]. It should be noted that this effect was manifested despite the fact that in CytC 8Mut, a substitution of the K72E residue within the red Ω-loop was introduced. Obviously, the replacement of one residue was not sufficient for the increase in the rigidity of the hemoporphyrin microenvironment observed in [41]. These data are consistent with the results of our study, indicating an increase in the tendency to particle aggregation and the lability of the Fe…S(Met80) bond, which may indicate a slight imbalance in the protein structure as a whole. The influence of the lability of the bond between heme iron and methionine sulfur on the ability of CytC to interact with external ligands and catalyze the peroxidase reaction will be discussed in more detail in the second article of this special issue.

## 5. Conclusions

The constructed and studied CytC 8Mut variant, featuring substitutions of five Lys residues in the UBS and three compensating Glu residues, is essentially devoid of electron transport function, as previously shown both in the mitoplast system [20] and in our experiments on the interaction with solubilized CcO. However, the mutant protein retains sufficient antioxidant activity against the superoxide anion radical (up to 50%). The introduction of site-directed mutations into the UBS CytC, along with the alteration of the dipole moment of the molecule, resulted in significant functional modifications in the mutant proteins, as well as certain changes in their physico-chemical properties; for instance, an increased aggregation tendency (2Mut and 5Mut), suggesting a potential destabilization of the mutant proteins’ structure. Interestingly, according to circular dichroism data, the secondary structure composition of mutant CytC 2Mut and 8Mut did not undergo substantial alterations. Consequently, the derived variant of CytC 8Mut, containing multiple mutations in the UBS, holds promise as a potential detector of the generation rate of superoxide anion radicals in mitochondria and other systems.

## Figures and Tables

**Figure 1 cells-12-02316-f001:**
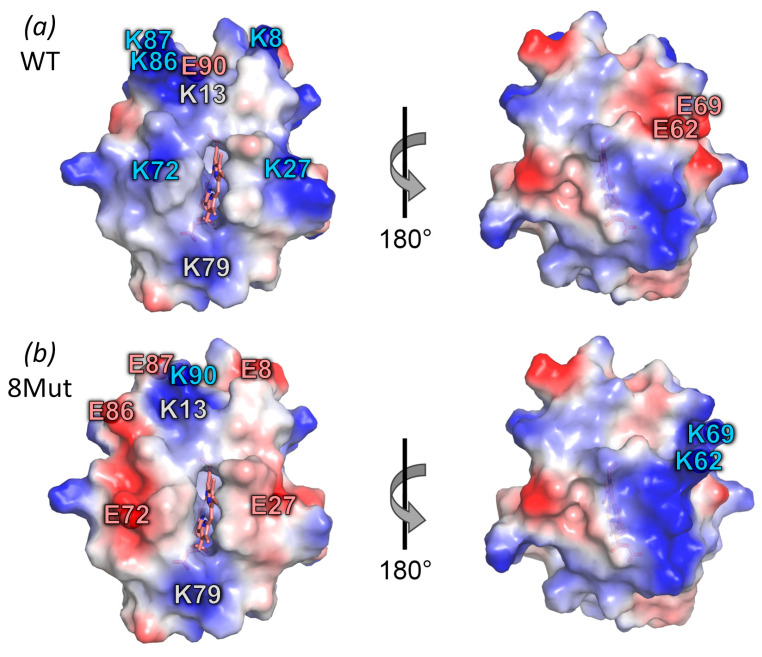
Locations of mutated residues in the 3D structure of CytC. Qualitative electrostatic potential surface of horse CytC WT (**a**) and 8Mut (**b**) are shown from the side of the heme cavity and when rotated by 180° around the vertical axis. ε-amino groups of Lys residues and carboxyl groups of Glu residues subjected to mutagenesis are marked with blue and red balls/numbers, respectively. The positions of unaltered Lys13 and Lys79 residues near heme cavity are marked with grey numbers. The images are obtained using the PyMOL program (pdb code 1HRC).

**Figure 2 cells-12-02316-f002:**
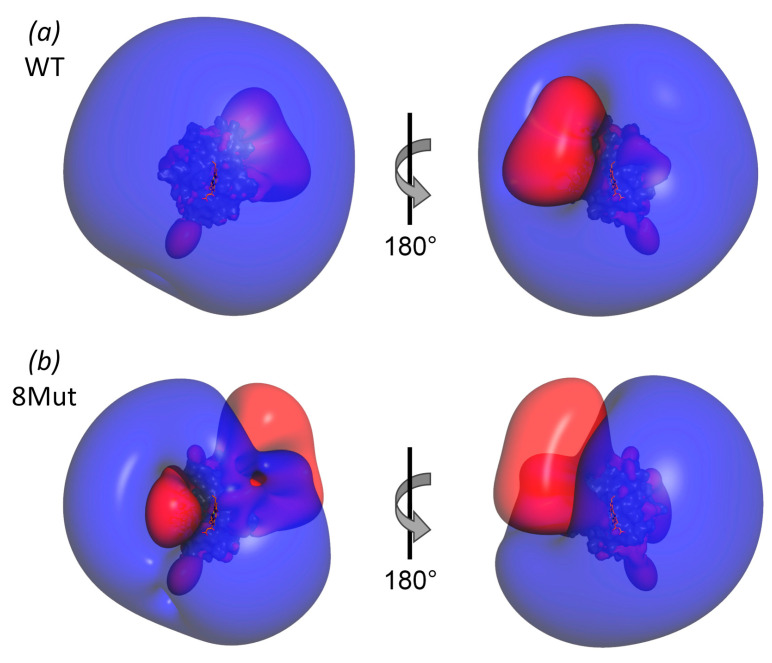
Isoelectric surfaces of CytC WT (**a**) and 8Mut (**b**). Blue and red surfaces correspond to charge +0.1 mV and −0.1 mV, respectively. Orientation of the molecules is the same as in Figure 1.

**Figure 3 cells-12-02316-f003:**
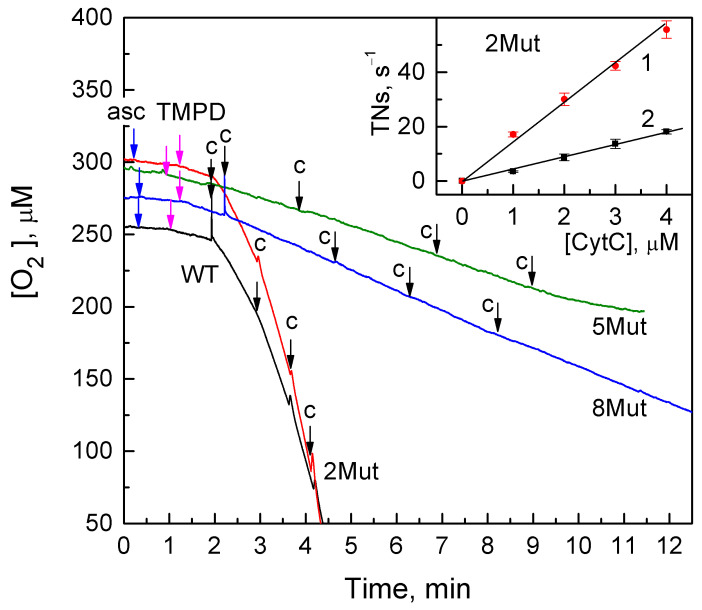
Typical oxygen consumption kinetics of CcO oxidizing CytC in the presence of TMPD and ascorbate: WT (black line) and CytC mutants: 2Mut (red line), 5Mut (green line), and 8Mut (blue line). The final concentrations in the measurement medium were CcO, 24 nM (black line) and 120 nM (red, green, and blue lines); ascorbate (5 mM); TMPD (0.1 mM). The addition of cytochromes (1 µM each) is marked with arrows with the letter c. The inset shows the dependences of the oxygen consumption rate on the concentration of CytC 2Mut, obtained in the presence of TMPD (1, red dots) and without it (2, black dots).

**Figure 4 cells-12-02316-f004:**
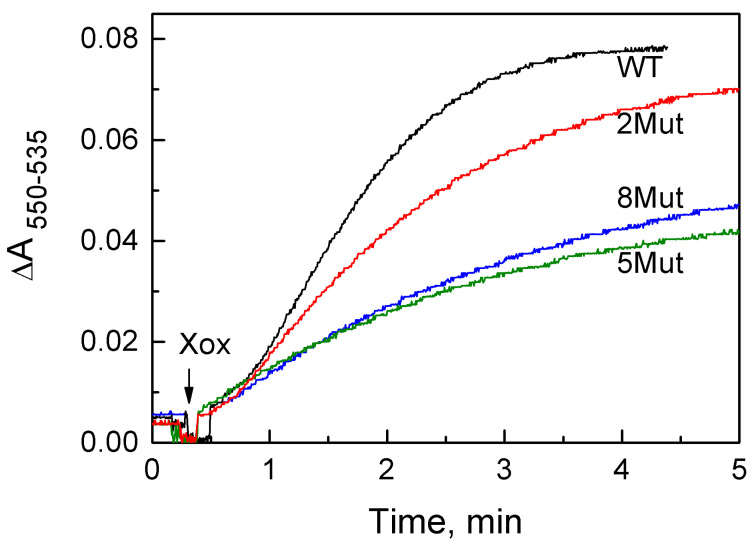
Typical kinetics of superoxide anion radical reduction during hypoxanthine oxidation by xanthine oxidase, measured by the reduction of CytC WT (black line) and mutant forms of CytC 2Mut (red line), CytC 5Mut (green line), and CytC 8Mut (blue line). CytC reduction was recorded spectrally in the two-wavelength mode by the difference in absorption at 550 nm and 535 nm (comparison wave). The concentration of CytC in the measurement medium in all experiments was 4 µM, hypoxanthine, 50 µM, and xanthine oxidase, 0.015 U/mL. The measurements were carried out in 50 mM K-phosphate buffer, pH 7.5, 0.1 mM EDTA.

**Figure 5 cells-12-02316-f005:**
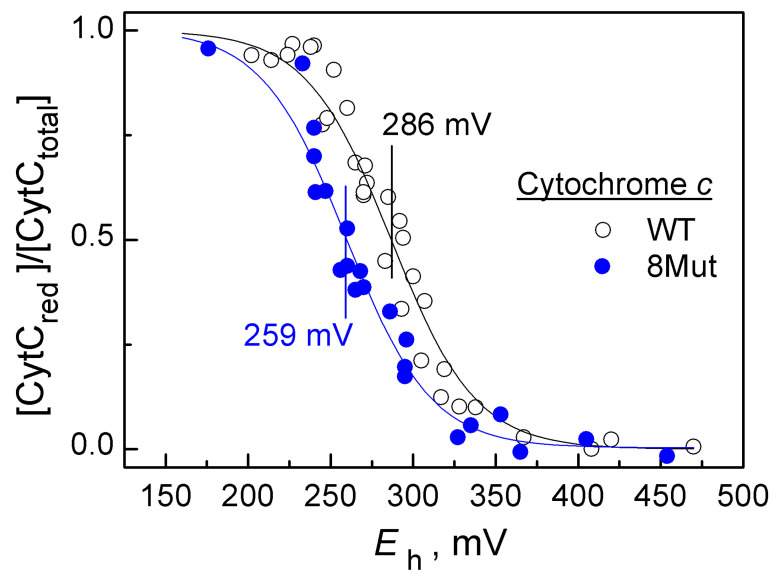
The redox titration curves of CytC, wild-type (WT—transparent circles), and mutated (8Mut—blue circles) forms.

**Figure 6 cells-12-02316-f006:**
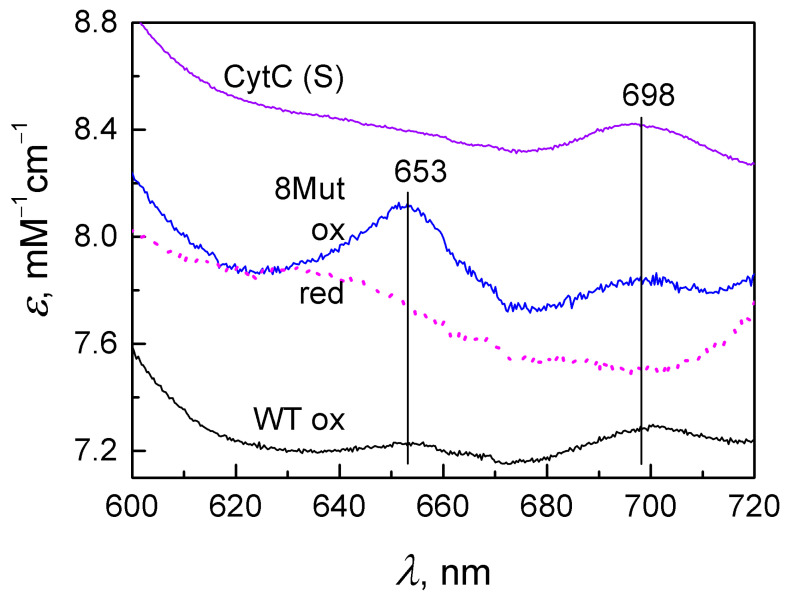
The longwave region of the absorption spectrum of ferric CytC: CytC (S) (magenta line), horse heart CytC (Sigma, Burlington, MA, USA), recombinant CytC WT (black line), and CytC 8Mut (blue line). For comparison, the spectrum of ferrous CytC 8Mut is also shown (red line, dotted line). Absorbance is normalized to the concentration of cytochromes in measurement samples that have been pre-oxidized with substoichiometric amounts of ferricyanide. The measurements were carried out in basic buffer, pH 7.6.

**Figure 7 cells-12-02316-f007:**
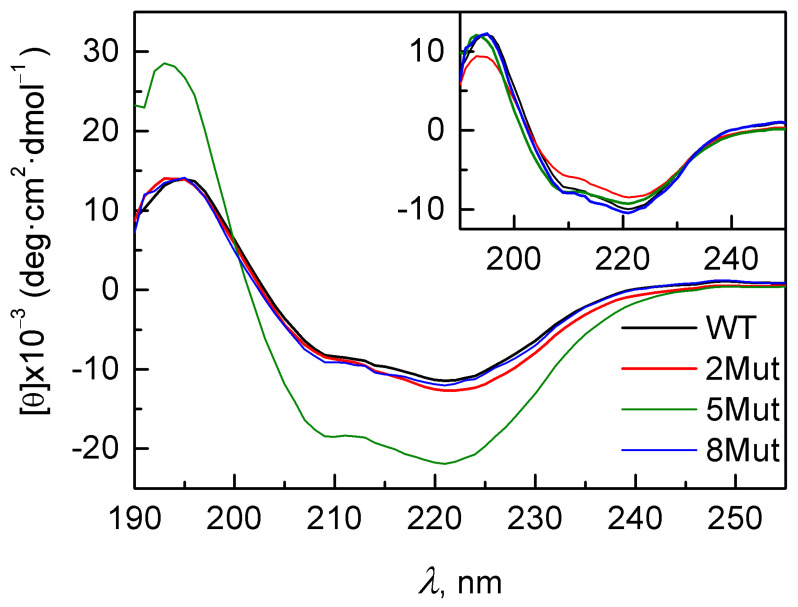
CD spectra of CytC WT (black line) and mutant variants 2Mut (red line), 5Mut (green line), and 8Mut (blue line). The spectra were normalized to the concentration of CytC in the studied samples, which was about 0.6 mM (measured from the band at 550 nm). The inset shows what the same CD spectra look like normalized to protein concentration based on far UV molar extinction at 205 nm, 210 nm, and 215 nm (Table 4 in [38]). Three values are averaged.

**Figure 8 cells-12-02316-f008:**
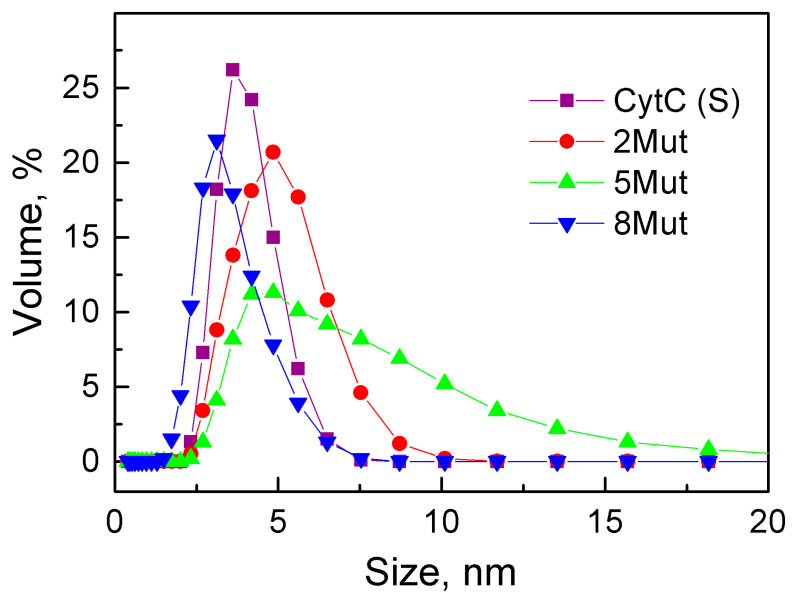
Particle size of CytC samples: CytC (S) (black squares), horse heart CytC from Sigma, and mutant forms 2Mut (red circles), 5Mut (green triangles), and 8Mut (blue triangles) measured by dynamic light scattering.

**Table 1 cells-12-02316-t001:** The activity of different CytC in the cytochrome *c* oxidase reaction.

Type of CytC	WT	2Mut	5Mut	8Mut
Activity, s^−1^	210 ± 10	54 ± 4	<1	<1
%	100	25	<1	<1

**Table 2 cells-12-02316-t002:** Reduction of different CytC by superoxide.

Type of CytC	WT	2Mut	5Mut	8Mut
Rate, uM O_2_^−^/min	1.63 ± 0.07	1.06 ± 0.17	0.72 ± 0.12	0.815 ± 0.09
%	100	65	45	50

**Table 3 cells-12-02316-t003:** The effect of amino acid substitutions on *E*_m_^0^ of CytC mutants.

Mutant	2Mut	5Mut	8Mut
Amino acid substitutions	E69K/K72E	K8E/E69K/K72E/K86E/K87E	K8E/K27E/K72E/K86E/K87E/E62K/E69K/E90K
Effect on *E*_m_^0^ (mV)	−26.2	−43.2	−48.1

**Table 4 cells-12-02316-t004:** The secondary structure analysis of cytochrome c (WT) and its mutant variants.

CytC	α-Helix, %	β-Strand, %	β-Turn, %	Disordered, %	NRMSD ^1^
WT	25.35	21.65	22.15	30.8	0.066
2Mut	31.05	16.9	21.3	30.85	0.066
5Mut	46.6	6.4	19.9	27.1	0.054
8Mut	28.4	20.3	21.8	29.6	0.073
S (Sigma)	22.45	24.3	22.15	31.2	0.078

^1^ The given values of normalized standard deviations (NRMSDs) are used as a statistical estimate of the difference between the experimental spectrum and the theoretical spectrum calculated on the basis of the obtained composition of the secondary structure. According to Kelly et al. [38], the NRMSD value in the calculations should be <0.1, which indicates a high reliability of the calculations.

## Data Availability

All raw data are available from corresponding authors under reasonable request.

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
