# Peer review of "Mutant Cytochrome C as a Potential Detector of Superoxide Generation: Effect of Mutations on the Function and Properties"

_cells, 2023, doi:10.3390/cells12182316_

Round 1

Reviewer 1 Report

In this manuscript Chertkova et al aim to characterise mutant forms of cytochrome c that are deficient in electron transport in the ETC but are still able to scavenge the superoxide radical. The group have previously published data on some of the mutants, including 8Mut for which they have previously shown loss of ETC activity while superoxide radical scavenging is retained. The same results were obtained in the submitted manuscript, using different methods. Therefore the advance made in this paper is minor.

The second paragraph of the Introduction claims that CytC has a relatively flexible structure associated with various functions but no references are provided to support these statements, and I am not convinced that this is a correct interpretation of the published information. 

The methods are described appropriately and the results are interpreted correctly. However exactly what new insights are provided is unclear. 

Some unusual phrasing is used in places, but the meaning remains clear. 

Author Response

We thank the Reviewer for the careful reading of the manuscript and valuable comments. With Reviewer’s permission, we give answers to comments step by step.

  1. The second paragraph of the Introduction claims that CytC has a relatively flexible structure associated with various functions but no references are provided to support these statements, and I am not convinced that this is a correct interpretation of the published information.

Answer. The Introduction contains links to several comprehensive review publications, for example, [7], [11], [13], describing the functional activities of CytC and its structural forms. Taking into account the relationship between the structure and function of the protein as a whole, as well as the presence of many functionally active forms of CytC, it seems to us that such interpretation is adequate.

  1. The methods are described appropriately and the results are interpreted correctly. However exactly what new insights are provided is unclear.

Answer. Our article presents previously unpublished data, including the construction of a CytC 8Mut variant with completely suppressed respiratory activity, during which the replacement of Lys residues by Glu was carried out in such a way as to shift the dipole moment of the cytochrome c molecule. The dipole moment plays an important role in orienting proteins in the correct directions, which increases their ability to bind to other partner molecules during the formation of a functional complex. In addition to the data obtained earlier on mitochondrial preparations, we have shown that, in addition to maintaining the main physicochemical characteristics, the CytC 8Mut mutant variant retained a sufficient ability to neutralize the superoxide radical also in systems that did not contain mitochondrial preparations.

Reviewer 2 Report

The subject of the paper is appropriate for the publication in the journal Cells. The submitted paper deals with the construction of several mutants of cytochrome c (cyt c) by substitutions of Lys residues in the universal binding site (UBS) in cyt c.  As the main results of the work can be considered fact that the authors have found variant cyt 8Mut with five Lys mutations in UBS which is essentially devoid of electron transfer function, however, retains its capability to oxidize superoxide anion radical. The secondary structure of this mutant is not substantially altered in comparison with the wild type of the enzyme.  Generally, the manuscript is written well with clear conclusions. However, before I can recommend this paper to be published, the authors should answer and clarify the following points:

1.     The first sentence of the Introduction should be corrected. More precisely would be “………ubiquinol-cytochrome c oxido-reductase (complex bc1) and cytochrome c oxidase (CcO) in the electron transfer chain (ETC) located in the inner mitochondrial membrane

2.     The meaning of the following expression is not clear “…and oxidation by CcO provides the redox control of the process from ETC” (p.3, l.93-94). The expression should be corrected or more clarified.

3.  The last paragraph of the section 3.1. “Mutations in the cyt c gene………..characterized previously” should be displaced to the Material and methods.

4.   Have the authors performed a control experiment about the reduction of cyt c (and all the mutant variants) in the presence of ascorbate and TMPD and absence of CcO ?

5.     From Fig. 4 it seems that not all 8Mut (as well as 5Mut) cyt c are reduced by superoxide anion (if the concentrations of cyt c variants are the same). What is the reason of this fact?  

6.     On page 9, l. 359-360 is written “..did not affect the rate of superoxide reduction”. More precisely would be “superoxide annihilation” or “superoxide oxidation”.

7.                       The paragraph 3.3 (p. 10-11) should be shortened.

8.       In the visible absorption spectrum of 8Mut cyt c a new band centered

      at 653 nm is observed, which is absent in the spectrum of WT cyt c. 

    Which transition is responsible for appearance of this band?

9.  I would recommend to use expression “dynamic light scattering” instead of “photodynamic light scattering”.

10.  It is noticed that WT cyt c has a tendency to form a dimer. Why is this property of WT cyt c different from the cyt c (S)? If I understand well, the composition of these two cyt c should be the same.

11.   In the last part of the Conclusion is suggested that variant Mut8 cyt c holds promises as a potential detector of the generation of superoxide anion radicals in mitochondria. How concretely would be this detection performed?

Several grammatical and stylistic errors are present in the manuscript. The English of the manuscript should be checked before the publication.

Recommendation: Before publication, the authors should address the above mentioned points.

Several grammatical and stylistic errors are present in the manuscript. The English of the manuscript should be checked before the publication.

Author Response

We thank the Reviewer for the careful reading of the manuscript and valuable comments. With Reviewer’s permission, we give answers to comments step by step.

  1. The first sentence of the Introduction should be corrected. More precisely would be “………ubiquinol-cytochrome c oxido-reductase (complex bc1) and cytochrome c oxidase (CcO) in the electron transfer chain (ETC) located in the inner mitochondrial membrane

Answer. Done.

  1. The meaning of the following expression is not clear “…and oxidation by CcO provides the redox control of the process from ETC” (p.3, l.93-94). The expression should be corrected or more clarified.

Answer. CcO is not only the terminal link of the respiratory chain providing energy for OXPHOS, but it also provides a runoff to oxygen for redox equivalents in the intermembrane space (IMS) of mitochondria with its elaborate protein folding redox machinery. Mia40 (mitochondrial import and assembly) protein, often referred as CHCHD4 in animals, is the central oxidoreductase of IMS disulfide relay. The enzyme is endowed with a highly redox active CPC (cysteine-proline-cysteine)-motif disulfide responsible for disulfide bond formation and oxidative protein folding in the mitochondrial IMS, the enzyme being active in the oxidized state only. And it means in turn that efficiency of CytC oxidation by CcO provides the redox control of the folding redox machinery from ETC. (Discussed in detail in the article by Vygodina et al BBA 2017 [16], which is referred to in the Introduction).

  1. The last paragraph of the section 3.1. “Mutations in the cyt c gene………..characterized previously” should be displaced to the Material and methods.

Answer. Done.

  1. Have the authors performed a control experiment about the reduction of cyt c (and all the mutant variants) in the presence of ascorbate and TMPD and absence of CcO?

Answer. CytC does not interact with oxygen and it is easy to reduce completely all mutant forms by 5 mM ascorbate and 0.1 mM TMPD. We obtained stable reduced forms of mutants using ascorbate+TMPD or dithionite with no difference.

  1. From Fig. 4 it seems that not all 8Mut (as well as 5Mut) cyt c are reduced by superoxide anion (if the concentrations of cyt c variants are the same). What is the reason of this fact?

Answer. It is obvious that the rate of reduction of CytC of 8Mut and 5Mut is slow and the reaction is far from complete thus it is not correct to approve incomplete reduction of these two forms.

  1. On page 9, l. 359-360 is written “did not affect the rate of superoxide reduction”. More precisely would be “superoxide annihilation” or “superoxide oxidation”.

Answer. Done.

  1. The paragraph 3.3 (p. 10-11) should be shortened.

Answer. Done.

  1. In the visible absorption spectrum of 8Mut cyt c a new band centered at 653 nm is observed, which is absent in the spectrum of WT cyt c.

Which transition is responsible for appearance of this band?

Answer. Yes, we found a new band but it is present in all recombinant cytochromes including WT only its magnitude is 3-4 times greater in the spectrum of 8Mut and is absent in commercial CytC from horse heart so we assumed it to be the result of some peculiarities of protein folding in E.coli cell. Nevertheless, we do not exclude that it reflects some transition but we present a study of the nature of this transition to other researchers as it was not the aim this study.

  1. I would recommend to use expression “dynamic light scattering” instead of “photodynamic light scattering”.

Answer. Done.

  1. It is noticed that WT cyt c has a tendency to form a dimer. Why is this property of WT cyt c different from the cyt c (S)? If I understand well, the composition of these two cyt c should be the same.

Answer. As mentioned above in the answer to question 8, folding of the recombinant CytC in E. coli cells may have a number of features compared to folding in animal cells. Therefore, we attribute the appearance of recombinant CytC WT dimers to our bacterial production system, in which overexpression of the target gene and a high protein yield are often observed during CytC WT biosynthesis. Apparently, under conditions of overexpression, some protein molecules form dimers during folding. The expression of mutant CytC genes, as a rule, does not reach such high levels as WT, and sometimes it is completely hindered, therefore, in their case, dimers are not observed. It should be noted that the presence of the dimeric fraction does not in any way affect the functionality of the recombinant CytC WT.

  1. In the last part of the Conclusionis suggested that variant Mut8 cyt c holds promises as a potential detector of the generation of superoxide anion radicals in mitochondria. How concretely would be this detection performed?

Answer. Wild-type СytC added to mitochondrial preparations (for example, to submitochondrial particles or mitoplasts - mitochondria lacking an outer membrane) is restored to a greater extent by components of the respiratory chain than by superoxide, and, therefore, cannot serve as a reliable agent for the detection of superoxide anion- mutant variants of cytochrome c, lacking respiratory function, but retaining antioxidant activity, are a potential basis for creating test systems for the determination of superoxide. Such a system can be effectively used, for example, to quantify the rate of superoxide generation by mitochondrial preparations under certain conditions.

Several grammatical and stylistic errors are present in the manuscript. The English of the manuscript should be checked before the publication.

Answer. English grammar was checked and corrected through the text.

Reviewer 3 Report

The manuscript by Chertkova et al. investigated the possibility of turning soluble cytochrome c into a superoxide scavenger. The material used is largely the mutants reported previously (reference 19). Given the deep involvement of this cytochrome c in  the apoptosis, the work comes with merits.  In addition, mechanistic study into the protein functional change is also of great value. This reviewer only has some minor comments.

1. The abstract is way too long. 

2. I would suggest to mention CcO in fact an oxygen reductase at least once because it is the correct name for the enzyme.

3. As the mutants used here are previously constructed and reported, the first section of the results should be greatly shortened. 

4. While the manuscript is written well, extra attention should be paid to avoid low-level errors. For example, line 20, bc1; line 35, In contrast to  CytC; lines 515-517, when compared two items, the comparative degree rather than the superlative degree should be used. Please proofread the text thoroughly.

Author Response

We thank the Reviewer for the careful reading of the manuscript and valuable comments. With Reviewer’s permission, we give answers to comments step by step.

  1. The abstract is way too long.

Answer. The abstract is shortened.

  1. I would suggest to mention CcO in fact an oxygen reductase at least once because it is the correct name for the enzyme.

Answer. Done. The Introduction gives the correct name "ferrocytochrome c:oxygen oxidoreductase" (the current classifier EC 7.1.1.9)

  1. As the mutants used here are previously constructed and reported, the first section of the results should be greatly shortened.

Answer. Indeed, an article was previously published that describes the results of a study in systems of mitoplasts and submitochondrial particles of mutant variants of CytC, including CytC 8Mut [19]. However, the principle of constructing this mutant variant was not described in [19] or elsewhere. Therefore, we found it possible to describe the construction in detail, as well as to accompany the description with Fig.1 and Fig2. However, the last paragraph of the section 3.1. “Mutations in the CytC gene………..” was displaced to the Material and methods.

  1. While the manuscript is written well, extra attention should be paid to avoid low-level errors. For example, line 20, bc1; line 35, In contrast to  CytC; lines 515-517, when compared two items, the comparative degree rather than the superlative degree should be used. Please proofread the text thoroughly.

Answer. Checked and corrected.